# One Hundred Candidate Genes and Their Roles in Drought and Salt Tolerance in Wheat

**DOI:** 10.3390/ijms22126378

**Published:** 2021-06-15

**Authors:** Ieva Urbanavičiūtė, Luca Bonfiglioli, Mario A. Pagnotta

**Affiliations:** Department of Agricultural and Forest Sciences, Tuscia University, via S. C. de Lellis, snc, 01100 Viterbo, Italy; ievaurbanaviciute@yahoo.com (I.U.); luca.bonf94@gmail.com (L.B.)

**Keywords:** osmotic adjustment, ionic, redox homeostasis, transcription factors, salt-responsive genes, genetic diversity, germplasm, cross-transferability

## Abstract

Drought and salinity are major constraints to agriculture. In this review, we present an overview of the global situation and the consequences of drought and salt stress connected to climatic changes. We provide a list of possible genetic resources as sources of resistance or tolerant traits, together with the previous studies that focused on transferring genes from the germplasm to cultivated varieties. We explained the morphological and physiological aspects connected to hydric stresses, described the mechanisms that induce tolerance, and discussed the results of the main studies. Finally, we described more than 100 genes associated with tolerance to hydric stresses in the *Triticeae*. These were divided in agreement with their main function into osmotic adjustment and ionic and redox homeostasis. The understanding of a given gene function and expression pattern according to hydric stress is particularly important for the efficient selection of new tolerant genotypes in classical breeding. For this reason, the current review provides a crucial reference for future studies on the mechanism involved in hydric stress tolerance and the use of these genes in mark assistance selection (MAS) to select the wheat germplasm to face the climatic changes.

## 1. Global Situation

Water and consequently an adequate hydric balance are essential for life, determining crop production in terms of quantity and quality. In the present century, the main problems are related to hydric stress, including dwindling water resources and excessive salinity of irrigation water and soil. Moreover, it is expected that global warming and climate change would increase the frequency of drought events, leading to a severe threat to food security [1,2,3].

In nature, plants are exposed to various abiotic factors and exhibit complex adaptation responses that depend on their degree of plasticity. In addition, they are facing environmental changes that increase the risk factors, such as rainfall distribution, loss of soil fertility and organic carbon, fluctuation in temperature and light, evaporative intensity, biological stresses, increasing pollution, and declining biodiversity [4]. Water is the most critical factor for the growth and development of living organisms, determining 82% of the variation in grain yield in areas receiving less than 400 mm of annual rainfall [5]. Land plants are anchored in the soil and rely on their root system to ensure water availability and nutrient uptake. About 30% of the Earth’s land is arid or semi-arid, and this percentage is expected to increase due to climate change [6]. The prevision indicates an erratic rainfall distribution, an overall decrease in total annual rainfall, and a more pronounced occurrence of dry periods [7], in addition to an increase in temperatures with a consequent reduction in ecosystem health through species extinction or a reduction in biodiversity, due to migration and changes in behavioral patterns [8].

Among all stresses, drought and salinity (soil and water) are the main abiotic factors that reduce a crop’s productivity, causing the most damage for grain production all over the world [9]. Other than grain production, water scarcity also affects the growth rate, leaf size, stem extension, root proliferation, susceptibility to disease, plant color, etc. [10,11]. Particularly in conventional farming, the rainfall water effectively decreases due to increased surface runoff, soil evaporation through unplanted and bare soil surfaces, and evaporation caused by the aeration of soil during intensive tillage [3]. Factors such as rainfall intensity, duration, exact time of rain events, and the soil properties affect the soil moisture status, especially the plant-available water (PAW), which can differ significantly with the same total amount of precipitation under different conditions. In addition, the water availability for plants decreases due to the high soluble salt concentrations in the soil, which causes absorption difficulties; i.e., water deficit or hydric stress. The main causes of soil salinization can be classified as natural or caused by human activity, named primary and secondary salinization, respectively (Figure 1) [12].

Current estimations indicate that salt issues affect about 20% of global lands and almost 40% of irrigated lands. This phenomenon is increasing and 50% of total cultivated land worldwide will be salinized by 2050 [13,14,15]. Global warming is the increase in temperatures often associated with rainfall decrease, leading to more extreme and unpredictable events. As a conclusion, the need to have a system protecting agriculture from an erratic climate becomes important worldwide.

The global grain production is an important indicator of food security, but unfortunately it grows slower than the human population. This unfavorable situation is exacerbated by abiotic factors such as drought, high salinity, extreme temperatures, flooding, and heavy metals [16,17]. There is an urgent need to increase food, which cannot be done by simply raising the agricultural lands, which causes deforestation and consequently climate change problem. Increasing the production per unit area (i.e., yields) represents a potential solution. Yields can be improved by using genetics methods, appropriate crop rotation, adequate fertilization, and early planting. For example, a six-week delay in planting time resulted in yield reductions of about 42% in barley and 22–32% in wheat [18]. While, early planting, proper fertilization, and appropriate crop rotation were found to increase the water-use efficiency (WUE) [19].

Useful and adequate breeding programs, addressed to reduce crop yield lost under stress conditions, are necessary to face the consequences of climate change. Attention should be focused on the drought-related traits, which include leaf canopy temperature, cell osmotic adjustment, cell membrane stability, leaf water potential, stomatal resistance, leaf rolling index, and leaf waxiness [18]. In wheat, for example, the total aboveground part of the plant (biological yield) in a water-limited production area is directly related to the water supply [20]. Although biological yield offers little hope of modification through breeding, the harvest index, which reflects the ratio of grain to total biological yield, appears to be amenable to genetic improvement for enhanced drought tolerance.

For this reason, hereafter is reported an overview of the sources of genes presents in wheat and in its wild relatives, and the morphological and physiological mechanisms involved in their tolerance. Moreover, this review provides updated information on more than one hundred candidate genes that provide plasticity in wheat tolerance to hydric stresses, providing tools that open possibilities for breeding new tolerant wheat genotypes.

## 2. Germplasm

Significant differences in drought and salinity stresses were reported between plant species [21]. Their growth rates were highly different under stress, indicating that germplasm is the main reserve of useful genes that could be used to improve cultivated crops. Figure 2 shows the differences in salt sensitivity among some plant species. Tall wheat grass (*Thinopyrum ponticum* (Podp.) Z.-W. Liu and R.-C. Wang; (synonymous *Agropyron elongatum* (Host) P. Beauv.) and saltbush (*Atriplex anicola* Paul G. Wilson) are extremely tolerant to salinity. *Arabidopsis thaliana* (L.) is more sensitive to salinity while alfalfa (*Medicago sativa* L.) is much more tolerant, compared to cereals. Among the cereals, rice (*Oryza sativa* L.) is the most sensitive while barley (*Hordeum vulgare* L.) is the most tolerant. Within the intermediate tolerant species, durum wheat (*Triticum turgidum* subsp. *durum* (Desf.) Husn) is more sensitive than bread wheat (*Triticum aestivum* L.) [21].

### 2.1. Sources of Resistances in Cultivated Species

Genes to improve tolerance to drought and salt stresses could be available in accessions, especially landraces of durum and bread wheat, but also in their wild relatives. Landraces have been successfully used to obtain salt-tolerant varieties by specific breeding programs [22,23]. The tolerance to these stresses could be measured with diverse methodology and observations. Some of these methods focus on the tolerance mechanisms, such as the number and dimension of stomata, or on the evaluation of the Na^+^ and K^+^ ion concentrations in leaves, or on looking at the root system and architecture. A more empirical tolerance evaluation looks at the differences in yield under stress levels. This latter approach is more useful for direct utilization and study of the effect on agricultural production. In turn, the former give information about single genes, information useful for gene pyramiding and to understand the physiological mechanism involved in the tolerance.

The homology among genomes is also different: the A, B, and D genomes contain similar loci that in some cases are difficult to be discriminated or to identify the specific locus for a specific function. In addition, the variability in terms of allele number depends of the species population. For example, the *Dreb* genes, one of the most important gene families conferring drought tolerance, are distributed in the different genomes. However, few alleles have been found related to abiotic stresses, and some allele-specific markers are developed for marker-assisted selection [24,25,26]. Within durum wheat, some alleles due to SNPs and/or INDEL mutations have been identified among varieties for *DREB1*, *HKT1*, and *WRKY1* genes [27,28]. In spite of the homology among the A, B, and D genome, some higher resistance to salt stress is present in the D genome. Bread wheat, *Triticum aestivum* L. (genomes AABBDD), accumulates less sodium and more potassium in expanding and young leaves than durum wheat, *T. turgidum* subsp. *durum* (Desf.) Husn (genomes AABB) [29].

### 2.2. Sources of Resistances in Alien Species

Wild relatives are useful sources of genes for stress tolerance. They could be used in breeding programs to transfer a single gene to reduce crop yield loss under stress (see below). The genes conferring resistance were transferred to the amphiploid, which often has very low quality and productivity. This is because the wild relatives, even if having genes able to confer resistance, are lacking in all the other genes selected during 10,000 years of agriculture and breeding, such as to have high yield and high-quality production. Several works [30,31,32,33] focus on the source or resistance available in several genomes of the *Triticeae* family, from the cultivated species to the donors of the A, B, and D genomes arriving to more distant species having genomes S, C, G, M, N, U, E, and J, up to the *Hordeum* genera (genomes I, H, and X). Table 1 reports a synthesis of the source of resistance found.

The transferability from the wild relatives to the cultivated ones is certainly different and depends on the genetic distance among species. Therefore, wild emmer, *Triticum dicoccoides* (Körn. ex Asch. and Graebn.) Schweinf (synonymous *Triticum turgidum* subsp. *dicoccoides*), is a useful potential donor for salt-tolerant genes, which can be transferred to cultivated cultivars by classical and/or modern techniques; while, to transfer genes from other more distant species, it would require passing through the amphidiploids. As reported in Table 1, the positive aspect is that there exists variability in salt tolerance amongst members of the *Triticeae*, with the tribe even containing a number of halophytes. Unfortunately, the amphiploid combines in the same genotypes the donors’ characteristics, which, together with useful tolerance genes, have also several unproductive characteristics. For this reason, few salt-tolerant varieties have been released from previous attempts at employing this approach. Modern technologies for assisted evolution give the possibility to transfer only the useful genes without transferring the whole background of the donor accession.

Gorham and coworkers analyzed several species and, looking at the tolerance of *Aegilops tauschii* (Coss.) Schmal (genome DD), which store less Na^+^, postulated the hypothesis that the D genome has the highest involvement in conferring tolerance [34,35]. Nevertheless, Na^+^ exclusion should not be the only mechanism since some tolerant individuals were found with a high level of sodium stored [36]. The *Kna*1 locus on chromosome 4D regulated the ratio of accumulated K^+^/Na^+^ to leaves under salt stress, for this reason bread wheat is generally a better Na^+^ ‘excluder’ than durum wheat. To improve durum wheat, the *Kna*1 locus was transferred from the D genome of hexaploid wheat into tetraploid wheat [29,37]. In the background of the durum cultivar Cappelli, the distal part of the long arm of chromosome 4B was substituted with chromosome 4D, creating some new germplasm with an enhanced K^+^/Na^+^ ratio but with not significantly different Na^+^ concentrations [37,38]; also, the A genome should have greater Na^+^ exclusion and K^+^/Na^+^ discrimination than the B genome, since *Triticum urartu* Thumanjan ex Gandilyan (AA) is more tolerant than durum wheat (AABB) [30]. This could be because the B genome negatively interact with the A genome, or because the B genome carry genes increasing Na^+^ entry [30]. Furthermore, within the tetraploid species there are accessions more tolerant to salinity than durum wheat; for example, wild emmer wheat (*T. turgidum* L. ssp. *dicoccoides* (Körn. Ex Asch. and Graebn.) Thell.) has lower rates of Na^+^ [39,40]. In durum wheat, the locus *Nax1* mapped on chromosome 2A has considerable variation in capacity to ‘exclude’ Na^+^ [41].

**Table 1 ijms-22-06378-t001:** Summary of the studies on a source of genes for tolerance to drought and salt, modified from [30,31].

Species	Genome	Common Name	Reference
*Triticum monococcum* L. ssp. *aegilopoides* (Link) Thell.	AA	Wild einkorn	[22,42]
*Triticum monococcum* L. ssp. *monococcum*	AA	Einkorn	[43]
*Triticum urartu* Thumanjan ex Gandilyan	AA		[42,44,45]
*Triticum turgidum* L. ssp. *dicoccoides* (Körn. Ex Asch. and Graebn.) Thell.	AABB	Wild emmer	[33,40,44,45,46]
*Triticum turgidum* ssp. *durum* L. (Desf.) Husn	AABB	Durum wheat	[22,47]
*Triticum aestivum* L. ssp. *aestivum*	AABBDD	Bread wheat	[48,49,50]
*Aegilops markgrafii* (Greuter) K. Hammer	CC		[34]
*Aegilops cylindrica* Host	CCDD	Jointed goat grass	[34,51,52]
*Aegilops triuncialis* L.	C^u^C^u^CC	Barb goat grass	[51]
*Aegilops tauschii* Coss.	DD	Goat grass	[34,36,45,53]
*Elytrigia elongata* Host Nevski	E^b^E^b^	Tall wheatgrass	[54,55,56,57]
*Thinopyrum ponticum* (Podp.) Barkworth and DR Dewey	EEEEEEEEEE		[55,58,59]
*Thinopyrum bessarabicum* (Savul and Rayss) Á. Löve	E^j^E^j^	Tall wheatgrass	[60]
*Triticum timopheevii* (Zhuk.) Zhuk. ssp. *armeniacum* (Jakubz.) Slageren	GGAA		[35]
*Triticum timopheevii* (Zhuk.) Zhuk. ssp. *timopheevii*	GGAA		
*Aegilops bicornis* (Forssk.) Jaub. and Spach	S^b^S^b^		[34,51]
*Aegilops sharonensis* Eig	S^j^S^j^		[34,51]
*Aegilops longissima* Schweinf. and Muschl.	S^j^S^j^		[34,51]
*Aegilops speltoides* Tausch var. *speltoides*	SS		[45]
*Aegilops searsii* Feldman and Kislev ex K. Hammer	S^S^S^S^		[34]
*Aegilops umbellulata* Zhuk.	UU	Jointed goat grass	[34,51]
*Aegilops biuncialis* Vis.	UUMM		[34]
*Aegilops ovata* auct.	UUMM	Ovate goat grass	[34,51]
*Aegilops variabilis* Eig	UUSS		[34,51]
*Thinopyrum junceiforme* (Á. Löve and D. Löve) Á. Löve	J_1_J_1_J_2_J_2_		[61]
*Thinopyrum scirpeum* (K Presl) DR Dewey	JJJJ		[61]
*Thinopyrum junceum* L. (Á. Löve)	JJJJEE	Sand couch Sea wheatgrass	[62]
*Aegilops comosa* Sm.	MM		[34]

Farooq and coworkers analyzed tolerance in *Triticeae* relatives. They specifically assessed the *Aegilopes* diversity working in Pakistan, which has about 6.8 million hectares of salty land [32,52,63]. Leaf ion concentrations was detected in several *Aegilops* species exposed to 50 mM of NaCl and the most tolerant and promising sources of tolerance were found in *Aegilops cylindrica* Host and in *Ae. geniculata* Roth. [60,61]. Although Cl did not differ significantly in any of the studied *Aegilops* species, they have differences in the Na^+^ and K^+^ concentrations in leaves. *Ae. comosa* Sm. and *Ae. umbellulata* Zhuk. (M and U genomes, respectively) had accessions with low Na^+^ concentrations. Several accessions of *Ae. tauschii* (DD), *Ae. cylindrica* (CCDD), and *Ae. ovata* auct. (UUMM) survived under severe stress [64].

The *DRF1* (Dehydration Responsive Factor 1) was isolated and characterized in *Ae. speltoides.* The *DRF1* belongs to the *DREB* gene family and encodes transcription factors playing a key role in water-stress responses. Studying the variation in *DRF1*, Thiyagarajan et al. [65] found a high similarity between the B and S genome, but also suggested that the two genomes have evolved independently.

*Thinopyrum bessarabicum* (Savul and Rayss) Á. Löve (JJ syn, EbEb; 2*n* = 14) could be much more convenient than polyploids for wheat cytogenetic manipulations. *Elytrigia. elongata* Host Nevski (EE; syn. *Lophopyrum elongatum* (Host) D.R. Dewey), the diploid tall wheatgrass, grows in salt marshes around the Mediterranean and survived exposure to 500 mM NaCl [55,66]. The hydric stresses tolerance is often different regarding the type of stress and the mechanisms of resistance. For example, *Ae. tauschii* is more drought-tolerant, while *E. elongata* is more salt-tolerant than other species in the *Triticeae* [54,67]. In addition, *Dasypyrum villosum* (L.) Borbás could be used as a source of resistance carrying, on chromosomes 5 and 6, loci with significant positive effects on salt tolerance [66]. To transfer tolerance from wild relatives to the cultivated species several interspecific crosses have been performed. *T. timopheevii* (Zhuk.) Zhuk. ssp. *timopheevii* (GGAA) has been hybridized with *Ae. tauschii* (DD) to make the synthetic hexaploid (GGAADD) [68,69]. Synthetic hexaploids have been made crossing durum wheat (AABB) and *Triticum monococcum* L. ssp. *monococcum*, *T. urartu*, and *T. monococcum* ssp. *aegilopoides* (Link) Thell. [68,69]. The aim was to have lines able to exclude Na^+^ and a higher K^+^/Na^+^ ratio.

## 3. Morphological and Physiological Response

When plants suffer from hydric stress, significant changes in their morphology can be observed. Usually, hydric stresses affect plant size, which becomes smaller due to a decreasing number of leaves and decreasing area. However, the roots are lengthened searching for water, which results in an increase in the root-to-shoot ratio. This adverse effect of water scarcity on crop plants causes fresh and dry biomass losses. Early maturity or early flowering is another adaptation strategy where a shorter vegetative phase in wheat can help to avoid stress in further very sensitive flowering and post-anthesis grain filling stages [70].

Drought induces a plethora of negative physiological alterations, such as cell turgor loss, reduction in CO_2_ assimilation, oxidative stress, and nutritional imbalance [63]. Nutritional imbalances due to drought stress occur by decreasing the water uptake and leaf transpiration, combined with alterations in nutrient uptake. Plants try to counteract these effects by activating drought resistance mechanisms, such as accumulation of salts and water, to improve the cell osmotic adjustment and stomata function. K^+^ and Cl^−^ ion regulation, as well water transport, have been important mechanism for plants under drought. K^+^ and Cl^−^ are involved in leaf cells osmotic adjustment, with the consequent water retention in cells; stomatal closure prevents water loss [71]. The presence of high ionic concentrations in soil, especially NaCl, is a great challenge for the plant’s physiology. To balance the high sodium concentrations in the soil, plants cells must maintain high concentration of potassium and low concentration of sodium inside the cells.

The root system has been studied less than the aboveground morphology. During the juvenile phases, the root architecture is plastic and can adapt to several environmental conditions. It has been detected that roots can escape a high salinity zone, moving to less salty soil. Galvan-Ampudia et al. [72] identified halotropism as the plant’s possibility to reduce their exposure to salinity by moving their root system to an adequate saline environment. The auxin distribution in the root tip affects the root response to salty environments, as summarized in Figure 3 [72]. Furthermore, changes in the PIN subcellular localization affect root auxin transport and can cause various deformities in root architecture, including the size, less lateral roots, root meristem collapse, and others [73].

Salinity inhibits the root length and reduces the number of roots, but also influences the root growth direction. Sun et al. [74] found that the root direction could change with NaCl increasing; the curved roots could be visible already with a concentration of 100 mM of NaCl in the media. Salt stress reduces gravitropism of root growth, altering the PIN-FORMED2 (PIN2) protein abundance and polar distribution and overly salt-sensitive (*sos1* and *sos2*) genes. Julkowska et al. [75] found that more than 100 genetic loci activated under salt stress are associated with root system architecture changes. Among these, CYP79B2 is correlated with lateral root development under salt stress while HKT1 reduce the lateral root development.

## 4. Mechanisms of Tolerance

Since plants cannot easily change habitats where conditions are more favorable, to survive and multiply, they evolutionarily adapted, gaining plasticity and specific responses. Generally, plant reactions to abiotic factors include morphological, physiological, biochemical responses, and various adaptation scenarios. In addition, a plant’s response can be either common to many kinds of stresses or specific to each one. Stress-tolerance mechanisms are divided in two macro groups: mechanisms involved in osmotic-stress tolerance and mechanisms involved in ion-stress tolerance [76]. The stomatal adjustment belongs to the first class of mechanisms. Plants under drought and osmotic stresses in non-irrigated systems tend to close their stomata to minimize transpiration, which is reflected in a reduction in growth and production. Exclusion and compartmentalization of toxic ions, avoiding their high concentration in plant cells, are mechanisms of ion-stress tolerance [21]. Each type of physical stress is translated by the plant into a biochemical response, and then into a pathway of interconnected signals with the expression of specific genes that are involved in the stress-tolerance mechanisms. Table 2 lists some of the genes that are candidates for the pathway process and their defense mechanisms for osmotic and ion stresses [21].

### 4.1. Osmotic Adjustment

Plants have an initial general response to stress by abscisic acid (ABA) biosynthesis in roots as a signaling molecule due to interactions with different receptors; it may activate the general or very specific response to a particular type of stress [85]. After ABA, the first plant-specific response to hydric stresses is to maintain the cell water content, by increasing osmotic force, named osmotic adjustment (OA), using various mechanisms described by Blum [86]. Molecules involved in osmotic adjustment signaling pathways regulate tolerance against osmotic stress via control of stomatal conduction, with the consequent accumulation of organic solutes in the cytoplasm, in order to decrease the osmotic potential of the cytosol.

The main mechanisms of defense against osmotic stress predict the accumulation of useful solutes and the use of polyamines. Among the first are proline, glycine betaine, sugars, and polyols.The proline content increases under salt stress at the intracellular level and acts as a reserve of organic nitrogen during the stress period. Deivanai et al. [87] highlighted how rice treated with proline improves its response under salt stress.Glycine betaine, known also as trimethyl glycine (TMG), is a quaternary ammonium compound with three methyl groups derived from glycine found in many plants and microbes. The TMG is electrically neutral on a wide range of pH and highly water-soluble, but it also contains groups of non-polar methylins. Due to its unique structural characteristics, it interacts with both hydrophobic domains and hydrophilic macromolecules, such as enzymes and protein complexes. Glycine betaine increases the osmolarity of the cell during the period of stress [88], stabilizes the proteins [89], protects the photosynthetic apparatus from stress damage [90], and then plays an important role in stress mitigation [91].Sugars: Plants under saline stress tend to accumulate carbohydrates that play a role in osmo-protection and energy reserves during the stress phases [92].Polyols are chemical compounds composed of multiple oxydrilic groups available for organic reactions. They are classified into two types: cyclical (e.g., pinitol) and acyclic (e.g., mannitol). Polyols acts as protectors or enzyme stabilizers when stress related to dehydration occurs [93].

The polyamines have a low molecular weight and are widely spread in the plant kingdom, playing various roles such as the regulation of somatic embryogenesis, cell differentiation, morphogenesis, seed germination, the development of flowers, and fruits and their senescence. Polyamines are associated with gene regulation processes, which are involved in the synthesis of solutes capable to maintain cell membrane integrity and related to the accumulation of Na^+^ and Cl^−^ ions [94].

### 4.2. Ionic Homeostasis

The response reaction as ionic homeostasis mainly focuses on the exclusion of the Na^+^ ion, avoiding toxic concentrations in plant tissue [21]. Plants can avoid the harmful effect of ions by reducing their accumulation in shoots or by transferring them into vacuoles [95]. One of the responses to ion stress was identified in the SOS (Salt Overly Sensitive) stress signaling and tolerance pathway. It has been shown that *Arabidopsis thaliana* respond to salt mainly through the SOS signal path, which consists of three components involved in ionic homeostasis. SOS1 encodes a plasma membrane Na^+^/H^+^ “antiporter” protein (i.e., a protein involved in the secondary active transport of different molecules through the membrane), which plays a critical role in sodium extrusion [96]. SOS2 encodes a kinase protein [97], while SOS3 encodes a Ca^2+^-binding protein that acts as a calcium sensor for salt tolerance [98].

Salinity produces two independent types of secondary responses in plants. The limited plant growth could depend on the osmotic effect of salt in the soil or to the accumulation of sodium ions within plant cells. Generally, osmotic stress has an immediate and great effect with a decrease in the growth of new shoots. Ionic stress acts secondly with milder effects on plants and only in environments with a high salinity rate, or on species highly sensitive to it, increasing the senescence of older leaves [21]. Munns [99] explained how hydric stress acts over time. As reported in Table 3, after a few minutes and up to a few hours after NaCl was imposed, plants undergo osmotic stress that slows down the growth rate of the leaves and roots, as the salt tends to seize the available Ca^2+^, seriously compromising root growth.

Rodriguez et al. [100] revealed that root growth was not compromised if Ca^2+^ was also administered along with NaCl. After a few days under saline stress conditions, the plants encounter ion stress, due to the accumulation of sodium ions and/or chlorine ions in the leaves’ cells, thus compromising mainly the epigea part of the plants. After weeks or months under saline stress, plants reduce the number and size of the new shoots, and the older ones start to yellow until death. It seems that only plants with an inner capacity to produce more shoots have a better chance to be maintained under adverse conditions, as they still have sufficient photosynthesis tissues. However, after weeks and months under stress, plants may also face death before the maturation of the seed. In cereals, it reduces the number of florets per ear, and alters the time of flowering and hence maturity [99].

### 4.3. Redox Homeostasis

Water shortages also cause an increase in the reactive oxygen species (ROS) levels and induce cellular redox homeostasis. Therefore, to avoid the harmful effects of these molecules, as a defense mechanism plants increase production of enzymatic and non-enzymatic antioxidants. The genes responsible for antioxidant activity are very important as candidates for breeding programs to increase abiotic stress tolerance. In plants under drought and/or saline stresses, deregulation or even disruption of the electron transport chain in chloroplasts and mitochondria can occur. This involves the formation of oxidizing molecules because oxygen acts as an electron acceptor, creating compounds such as hydrogen peroxide, radical oxydrile, or radical superoxide, which can damage cellular integrity [101]. Plants can defend against the ion stress with antioxidant molecules, including among the most important nitric oxide (NO) [96]. NO not only is involved in the regulation of various processes of plant growth and development, but it also reacts with free lipid radicals, protecting the oxidation of lipids. Furthermore, the NO helps in the activation of antioxidant enzymes, such as superoxide dismutase (SOD), catalase (CAT), glutathione peroxidase (GPX), ascorbate peroxidase (APX), and glutathione reductase (GR) [102].

## 5. Genes and Transcription Factors Involved

State-of-the-art molecular genetic approaches have provided valuable insights about the plant’s response to environmental change, and signaling pathways to activate mechanisms of adaptation. Moreover, identification of several genes associated with stress adaptation has led to rational breeding programs.

Hereafter, more than 100 genes associated with tolerance to hydric stresses in the *Triticeae* sp. are described. They are presented in the Appendix A, together with characteristics such as Accession Nr., dimension (bp), Annotation, Function, Primers, and reference. These genes belong to several groups of responses with different biological functions, and for easier discussion, we grouped them into osmotic adjustment (Appendix A), ionic (Appendix A), and redox homeostasis (Appendix A).

In plant cells, the adaptation to adverse conditions begins from signal perception to the production of functional proteins through activation of the target genes. Understanding of a given gene function, and expression pattern under hydric stress conditions, is particularly important for efficient selection of new tolerant genotypes in classical breeding.

### 5.1. Genes Involved in Hydric Stress Tolerance

Sixty-nine genes involved in osmotic adjustment (OA) are reported in Appendix A. These genes are responsible for defense mechanisms against osmotic stress via control of stomatal conductance; accumulation of organic solutes in the cytoplasm to decrease the osmotic potential of the cytosol; and ionic balance, i.e., the Na^+^/K^+^ ratio (the accumulation of sodium can be used for OA, especially if it is balanced by the accumulation of potassium). The initial general plant response to hydric stress is abscisic acid (ABA) biosynthesis in roots, which, as a signaling molecule due to interactions with different receptors, may activate the different mechanisms of tolerance [85].

The ABA phytohormone accumulation is one of the main and most common ways to inform plant tissues and cells about changes in environmental conditions. Therefore, genes involved in ABA synthesis play an essential role in the perception and transformation of information about adverse conditions. For example, the accumulation of ABA, and therefore increased expression of the *TaNCED1* gene isolated from *Triticum aestivum*, significantly improved drought tolerance in transgenic tobacco plants [103]. In addition, a previous study has shown that oxidative cleavage of cis-epoxycarotenoids catalyzed by NCED (9-cis-epoxycarotenoid dioxygenase) is the critical step in the ABA biosynthesis [104]. Phylogenetic data showed that *TaNCED1* has the highest (95%) identity with the barley *HvNCED1* gene [103]. The genes involved in the ABA biosynthetic pathway are potential candidates for breeding to improve plants tolerance to various stresses.

Furthermore, plants can transmit information about stress conditions through the lipid-dependent signaling pathway (Figure 4). Three phospholipases (A, C, and D) involved in this pathway produce secondary signaling messengers, which play important roles in plant response and tolerance under hydric stress [105]. Phospholipases D catalyze the hydrolysis of phospholipids (PC—phosphatidylcholine, and PE—phosphatidylethanolamine) to produce phosphatidic acid (PA), which is a very important second messenger that modulate many cellular processes, including the signaling pathways related to plant defense, and tolerance [106].

Another way to produce phosphatidic acid (PA) and transmit information about the environmental changes is through the regulation of the calcium ion concentration in the cytosol, via the phospholipid pathway [107]. The hydrolysis of phosphatidylinositol bisphosphate (PIP2) by phospholipase C produces diacylglycerol (DAG) and inositol triphosphate (InsP3). In animals, IP3 is responsible to release calcium, while in plants this function is related to IP3 and his derivative IP6 (Figure 4) [108,109].

Several studies reported that the *TaPLC1* gene is involved in plant response to cold, salt, and drought stresses. It encoded phospholipase C (PLC), a specific enzyme that mediates abscisic acid signal transduction in guard cells via increasing the cytoplasmic Ca^2+^ [110,111,112]. Moreover, it was found that a reduced level of phosphoinositide-specific phospholipase C in guard cells is associated with the inhibition of stomatal opening by ABA [112]. The importance of this gene for plant tolerance is demonstrated by its prevalence in the plant kingdom from GenBank: Arabidopsis *AtPLC1* (AT5G58670) common tobacco *NtPLC1* (AF223351.1), rice *OsPLC1* (AJ276277.2), cowpea (U85250.1), and *Lilium davidii LdPLC1* (AY735314.1).

Another pathway is the lipoxygenase-related (LOX) pathway and its metabolites, oxylipins and jasmonates, produced from free fatty acids (FFAs) that is released free by phospholipase A (PLA) from the membrane phospholipids (Figure 4). The LOX pathway based on polyunsaturated fatty acids oxidation, such as α-linolenic acid, is particularly important since it represents a substrate in the synthesis of the phytohormone jasmonic acid (JA).

Moreover, it has been demonstrated that biotic and abiotic stresses induce the genes involved in α-linolenic acid metabolism [105,113]. The first allene oxide cyclase (AOC) gene was isolated from *T. aestivum* cultivar SR3 (cross between Cv. JN177 and tall wheatgrass *Thinopyrum ponticum*). *TaAOC1* copy was 720 bp in length and was transcribed in all tissues even with different degrees. Zhao et al. [113] detected the highest levels of this gene at the booting stage, particularly in roots, but also in other aerial organs. Interestingly, during the anthesis, a high level of *TaAOC1* transcription was detected in the awn and ears [113]. Moreover, superoxide dismutase (SOD) activity was increased in transgenic plants constitutively expressing *TaAOC1*, indicating that the regulation of ROS homeostasis is activated by jasmonic acid (JA) to increase salinity tolerance.

As mentioned above, phospholipase C (PLC) activity releases calcium ions into the cytosol (Figure 4) and increases the amount of these ions, raising the interaction with “Ca^2+^ sensors families” such as calmodulins (CaM), CaM-like proteins, calcineurin B-like proteins (CBLs) and their interacting kinases (CIPKs), and Ca^2+^-dependent protein kinases (CDPKs) [114]. The Ca^2+^ ions bind with calcium-binding EF-hand proteins, which consist of a helix–loop–helix structure, and an interhelical loop of 12–14 amino acids that bind calcium ions. One of these calcium-binding EF-hand proteins—*TaCab1*—was isolated and characterized from wheat leaves, and is upregulated by biotic and abiotic stresses [115]. Moreover, high affinity calcium-binding proteins CRT (calreticulin protein), which was found in several plant species, is responsible for the hydric stress tolerance in crops [116]. In *T. aestivum*, a full-length cDNA 1446 bp was isolated, encoding calreticulin protein namely *TaCRT* [117], which has high homology with other plants’ CRT. Jia et al. [117] underline the role of *TaCRT* in drought tolerance, and found a higher water-use efficiency (WUE), water retention ability (WRA), relative water content (RWC), and lower membrane damaging ratio (MDR) under water-stress conditions. Besides, three *TaCRT* genes, named *TaCRT1*, *TaCRT2* and *TaCRT3-1,* have been identified in wheat. The all three genes were strongly induced under salt stress, but exhibited different expression patterns in different tissues. Their localizations were on 2 L, 5 L, and 3 L, respectively, with additional homologous copies in the three genomes A, B, and D [116]. *TaCRT1* with an open reading frame of 1287 bp is involved in defense responses and stresses tolerance in wheat. Moreover, Xiang et al. [116] detected an association between *TaCRT1* and the superoxide dismutase (SOD), peroxidase (POD), and catalase (CAT) activities, which suggest that they may be included in redox homeostasis pathways.

Calcium-binding proteins change their conformation after binding calcium ions, and this transformation leads to activation of various calcium-dependent protein kinases [114]. The SNF1-related protein kinase SnRK2 subfamily, belonging to Ser/Thr protein kinase class, has shown to be responsible for signal transductions in various stresses. Overexpression of the SnRK2 subfamily genes, such as *TaSnRK2.4*, *TaSnRK2.8*, and *W55a*, enhanced osmotic potential in transgenic Arabidopsis, and they can be used in breeding to improve osmotic adjustment in wheat species [118,119,120]. Subcellular localization showed that both *TaSnRK2.4* and *TaSnRK2.8* are presented in the cell membrane, cytoplasm, and nucleus, proving that they are responsible for signal transmission in plant cell. The most highly homologous in the GenBank library for these genes are members of the rice *SnRK2* subfamily: *W55a*-*OsSAPK1* (90.38%), *TaSnRK2.8*-*OsSAPK8* (94.8%), and *TaSnRK2.4*-*OsSPAK4* (92.5%).

Other protein kinases, such as *TaSK5* and *TaGSK1*, belong to the glycogen synthase kinase–shaggy kinases family, and they are involved in signal transmission in a stressful condition. The structure of these protein kinases contains both a Ser/Thr protein kinase catalytic domain and phosphorylation site, and they are signal transmitters in plant response to salt stress [121,122]. In common wheat, the Leucine-rich repeat receptor-like serine/threonine-protein kinase *TaER-B1* was strongly expressed, and upregulated by numerous environmental stresses, including drought and salinity [123].

Among the protein kinases *TaABC1* (Appendix A), a member of the ABC1 family was identified in *T. aestivum* with an activity on the BC1 complex. Members of the kinase family have in common a conserved domain, but they have differences that lead to different localizations of ABC1 proteins and probably cause differences in their functions. Wang et al. [124] studied the effects of *TaABC1* protein on transgenic Arabidopsis, underling its role in reducing water loss and increasing osmotic potential, photochemistry efficiency, and chlorophyll content. Moreover, *TaABC1* is related to the expression of *DREB1A*, *DREB2A*, *RD29A*, *ABF3*, *KIN1*, *CBF1*, LEA, and *P5CS*, which are well-known stress-responsive genes.

The plants require the expression of a large number of genes and specific transcription factor families (TFs) to activate their tolerance mechanisms. Among these, an important role is played by protein kinases, which use adenosine triphosphate (ATP) as a donor of the phosphate group for transcription factor phosphorylation (Figure 5). This mechanism has been identified as a plant response to abiotic stresses, which allows a fast plant response via fast switching transcription factors from the dephosphorylated state to phosphorylated state and back [125]. Together with protein kinases in TF activation phosphatases, ATP and/or ADP are involved.

According to the TFs’ DNA-binding domain characteristics, they were divided into multi-gene families (AP2/EREBP, bZIP, MYB/MYC, NAC, and WRKY) [125]. Moreover these families are phosphorylated by various protein kinases families (Figure 5), for example MYB/MYC and WRKY by mitogen-activated protein kinases (MAPKs), while NAC and DREB by serine-threonine kinases (SnRK2), and bZIP also by SnRK2 and some calcium-dependent protein kinases (CDPKs) [125].

The AP2/EREBP family has a domain structure divided into four subfamilies: AP2, RAV, ERF, and DREB. The AP2 transcriptional factor subfamily has two AP2 domains (RAV one AP2) and one B3 domain, only one AP2 domain characterizing DREB and the ethylene-response transcription factor (ERF) subfamilies [126]. The overexpression genes, such as *TaDREB1*, *TaDREB6*, and *WDREB2*, in transgenic plants enhanced the tolerance to abiotic stresses, such as drought, high salinity, and freezing, via the activation of stress-inducible genes (LEA/COR/DHN) with DRE/CRT cis-acting element in their promoter region [126,127,128].

The bZIP proteins belongs to the most large and diverse TF superfamily, classified into 14 subgroups, and carrying a highly conserved bZIP domain, which include a basic region and a leucine zipper. Members of bZIP family that carry ABA response element binding factors (AREBs) belong to subgroup-A, and are involved in the abscisic acid signaling response. These transcription factors regulate the expression of the genes responsible for drought tolerance and have the ABA-responsive cis-element (ABRE), as does numerous of the LEA family genes, such as the cold regulated (COR), responsive to dehydration (RD), early responsive to dehydration (ERD), and responsive to ABA (RAB) genes [129]. For example, in wheat bZIP, the gene *WABI5*, from group A, regulated the COR/LEA genes in stress responses such as freezing, osmotic, and salt stresses [130]. The other bZIP-type transcription factors from group S, such as *Wlip19* and *TaOBF1*, and their direct protein–protein interaction, also act as a transcriptional regulator of the COR/LEA genes for stress tolerance [131]. However, the bZIP transcription factor *TaABL1* regulates the expression of other LEA family genes, such as the osmotic adjustment-related function genes *RD29B* (responsive to dehydration), *RAB18* (responsive to ABA), and other stress-related genes, which control stomatal closure in transgenic Arabidopsis [132]. TFs from the subfamily of basic leucine zipper (bZIP) might be useful genetic resources for breeding tolerant genotypes.

One important and large subfamily of TFs MYB/MYC involved in response and tolerance to various stresses, and characterized by a DNA-binding domain, consist of helix–turn–helix (HTH) or basic helix–loop–helix (bHLH) domains [125]. The overexpression of MYB TFs, such as *TaMYB19-B*, *TaMYB56-B*, and *TaMYBsdu1*, cause the expression of a number of abiotic stress-related genes, and are important regulators involved in wheat adaptation to water scarcity [133,134,135]. A group of MYB genes respond to hydric stress (*TaMYB1*, *TaMYB29*, *TaMYB34*, *TaMYB57*, and *TaMYB72*) were isolated and identified in 60 wheat MYB genes during abiotic stress expression analyses [136]. In addition, the overexpression of this family members, such as *TaMYB32*, *TaMYB33*, and *TaMYB73*, which were specific salt-inducible genes, enhanced tolerance to salt stress in transgenic Arabidopsis [136,137,138]. Moreover, *TabHLH39*, as an MYC transcription factor, consists of the basic helix–loop–helix (bHLH) domain, and overexpression of this TF improved tolerance to drought, salt, and freezing in transgenic Arabidopsis [139]. MYB/MYC family proteins are crucial in plant stress responses and may improve crop tolerance to hydric stress.

A further large superfamily belongs to plant-specific WRKY transcription factors; its specific binding domain consists of a conserved WRKYGQK sequence followed by a zinc-finger motif. The importance of WRKY in abiotic stresses tolerance is shown via the regulation of some stress-responsive genes with specific cis-acting element W-box in their promoters, specific for the WRKY binding domain [140]. For example, *TaWRKY2* binds the promoter of gene *RD29B* (dehydration inducible gene), and *TaWRKY19* binds a few more, *RD29A* and *COR6.6*, which led to increased tolerance to water deficit in transgenic Arabidopsis [141]. Moreover, the overexpression of TFs *TaWRKY10* and *TaWRKY44* in transgenic tobacco plants significantly activated the genes involved in osmotic adjustment and ROS scavenging, improved tolerance to hydric stresses via proline and other soluble sugar accumulation, and enhanced the antioxidant system [142,143]. Besides that, WRKY TFs impart tolerance to abiotic stresses, they also control plant development and growth, such as *TaWRKY13* and *TaWRKY80*, which improved the root development characteristics together with an increased proline level in transgenic Arabidopsis [144,145].

Another plant-specific TF superfamily proteins feature is the NAC domain (for NAM, ATAF1/2, and CUC2) in the N-terminal region and various C-terminal sequences [125]. Multiple abiotic stresses in transgenic Arabidopsis cause overexpression of NAC TFs, such as *TaNAC2*, *TaNAC29*, *TaNAC47*, *TaNAC67*, and *TaNAC69*, which, in turn, improve dehydration tolerance via controlling expression of some abiotic stress-response genes [146,147,148,149,150]. Moreover, the NAC family members *TaNAC4* and *TaNAC8* overexpression in wheat were induced not only by abiotic stimuli but also by stripe rust pathogen infection. Both of them contained a plant-specific NAC domain in the N-terminus and transcriptional activity in the C-terminal region [151,152].

### 5.2. Stress Tolerance-Related Genes and Functional Proteins

The LEA3 proteins belong to the big “late embryogenesis abundant” proteins family, whose function is usually associated with maintaining the water content in plant cells under abiotic stresses [153]. These proteins, according to their structure, are characterized by an 11-mer amino acid motif (TAQAAKEKAGE), which are important in formatting protein α-helical during dehydration [154]. It was shown that transgenic *Phellodendron amurense* overexpressing wheat *TaLEA3* gene were tolerant to water deficit by fast stomatal closure [155]. Moreover, overexpressing LEA3 family genes, *WZY3-1* and *TaLEA3*, effectively enhanced the drought and salinity tolerance in transgenic plants due to protecting the cell membrane from damage, enhancing photosynthetic efficiency, and reducing ROS [156,157,158]. LEA3 family genes *Wrab18* and *Wrab19* encoded cold-responsive LEA/RAB-related COR proteins, which are induced under salt and drought stresses, too [159]. Both genes contain core ACGT motifs in the promoter regions and are direct target genes for b-ZIP type transcription factor *WLIP19* [131].

The late embryogenesis abundant proteins (LEA2) family, also named dehydrins, according to a combination of conserved segments, were subdivided into seven groups: KS, SK3, YSK2, Y2SK2, Kn, Y2SK3, and YSK3 [160]. SK3 dehydrin type genes *TaDHN2.1*, *TaDHN2.2*, and *TaDHN2.3* encode cold acclimation proteins, which protect the plasma membrane against freezing and dehydration stress via drought or salinity. Genes belonging to the YSK2 group encode dehydrins *TaDHN6*, *TaDHN11*, and *TaDHN17*, which their transcript levels were relatively higher only under dehydration stress. Moreover, dehydrins from the same group, *TaDHN9* and *TaDHN17*, induced expression under salt stress. Dehydration also induced expression of dehydrins from group Kn, such as *TaDHN18* and *TaDHN23*. Furthermore, expression of dehydrin *TaDHN7*, from the Y2SK2 group, was induced by both dehydration and salt [160]. Dehydrins genes are valuable for breeding programs; for example, overexpression of *DHN5* in transgenic plants increases tolerance to drought and salinity stress via osmotic adjustment [161].

Aquaporins belongs to another important group of functional proteins and it is responsible for regulating fast transmembrane water flow in plants. According to the amino acid sequence, homology, and protein subcellular localization, water-selective channel proteins are grouped into subfamilies, such as the nodulin 26-like intrinsic proteins (NIPs), the tonoplast intrinsic proteins (TIPs), the plasma membrane intrinsic proteins (PIPs), and the small basic intrinsic proteins [162]. The overexpression of the gene *TaNIP* (*Triticum aestivum* L. nodulin 26-like intrinsic protein) resulted in higher salt tolerance of transgenic Arabidopsis. Gao et al. [163] considered that the *TaNIP* gene induces plant salt tolerance mostly due to controlling the K^+^/Na^+^ ratio and Ca^2+^ concentrations, but not by proline accumulation. However, the *TaTIP2;2* (tonoplast intrinsic protein) overexpression reduced *P5CS1* expression and decreased the osmotic tolerance of transgenic Arabidopsis, due to the suppression of proline synthesis [164]. It has also been suggested that this gene may be a negative regulator of stress, due to the downregulation of *P5CS1* and other stress-tolerance-related genes [164]. Thus, aquaporin may be involved not only in fast water transport but also as a secondary function–intermediate regulator in processes related to the osmotic adjustment. For example, the PIP2 subgroup genes *TaAQP7* and *TaAQP8* provide drought and salinity tolerance in transgenic tobacco not only by controlling a better water state but also by inducing expression of antioxidant enzymes, which reduces the ROS levels and membrane damage [165,166]. Such a contrary response of the PIP subfamily genes to stress may be related to their subcellular localization. Osmotic adjustment is positively related to the *TaAQP7* and *TaAQP8* genes products, which are localized on the cell plasma membrane, while it is negatively related to *TaTIP2;* two products which are localized on the endomembrane system.

Among the many different mechanisms of defense against osmotic stress, there is the accumulation of useful solutes such as proline, glycine betaine, sugars, and others. The plants can regulate the amount of amino acid L-proline in two ways, either by increasing its synthesis or by inhibiting its degradation. Proline acts not only like an osmoprotectant, but it is also involved in stabilizing membranes, enzymes, cellular structures, and reducing reactive oxygen species (ROS) [167]. Therefore, genes involved in the proline synthesis, such as *P5CS* (pyrroline-5-carboxylate synthetase), *P5CR* (pyrroline-5-carboxylate reductase), and degradation *PDH* (pyrroline-5-carboxylate dehydrogenase), are very useful for drought tolerance and could help to enhance the yield of wheat under salinity stress [84].

### 5.3. The Specific Genes Involved in Ionic Homeostasis

Twenty-five genes involved in the Salt-Overly-Sensitive (SOS) signal transduction pathway are reported in Appendix A. These genes are responsible for the regulation of the ion transport system that facilitates ion homeostasis [168]. Calcineurin B-like (CBL) proteins act as a Ca^2+^ ion sensor and play an important role in signal transduction in response to various environmental stimuli. Some CBL family proteins (Appendix A) are responsible for very specific signaling pathways that ensure ionic homeostasis under salt stress. Moreover, a complex of CBL proteins with their protein-interacting protein kinases, CIPKs, have a crucial regulatory function in ionic homeostasis via the SOS salt stress-signaling pathway. At the beginning of this pathway, calcium-binding protein SOS3, activated by Ca^2+^ ions, formulate a complex with SOS2 (sucrose non-fermenting-related serine/threonine protein kinase), ready to activate the SOS1 gene. The SOS1 gene encodes a putative Na^+^/H^+^ anti-porter [169], which regulate the circulation of the Na^+^ ion from shoots to roots and accumulate it into the vacuoles [170,171]. Such CBL/CIPK complexes were identified in wheat, and the candidate gene for the calcium-binding protein SOS3 is *TaCBL4*, the orthologue of *AtSOS3*, which strongly interacted with six candidates genes for SOS2, TaCIPKs (*TaCIPK3*, *5*, *14*, *15*, *26*, and *31*) [77]. It was also found that other calcium-binding proteins, *TaCBL1* and *TaCBL9*, strongly interacted with five (*TaCIPK3*, *5*, *14*, *15*, and *25*) and two (*TaCIPK11* and *31*) CBL-interacting protein kinases, respectively [77]. In wheat, these complexes can activate H^+^/Na^+^ (*SOS1*) anti-porters, as plasma membrane *TaSOS1*, or vacuole *TNHX1* and *TaNHX2* [80,81,83].

In order to keep active the Na^+^/H^+^ anti-porters (*SOS1*), and remove the sodium ions into the vacuole in exchange for H^+^, the H^+^ gradient generated by the vacuole H^+^-ATPase and H^+^-pyrophosphatase (H^+^-PPiase) is crucial. In turn, it provides energy for successful exchanges. Among the three classes of H^+^-ATPase’s in plants, the vacuolar H^+^-ATPase (V-ATPase) is the most complex one, consisting of two sub-complexes: the peripheral *V*_1_, responsible for ATP hydrolysis, and the membrane-integral *V*_0_ complex, responsible for proton translocation. In wheat, each sub-complex consists of a number of subunits, whose overexpression evokes tolerance for salinity in transgenic Arabidopsis [172]; for example, subunits A, D, and G from the sub-complex *V*_1_, are responsible for ATP hydrolysis and H^+^ transport, and sub-units C, F, and H are responsible for the stabilization of sub-complex *V*_1_ and its connection with *V*_0_ [173]. As mentioned earlier, vacuole H^+^-pyrophosphatase (H^+^-PPiase) generates a proton gradient and controls Na^+^/H^+^ transport, but uses the energy for hydrolysis of pyrophosphate (PPi) molecules [80].

High-affinity K^+^ transporters (HKT) belong to another gene family that helps regulate potassium transportation, but that are not involved in the SOS pathway. They are common in many plant species, including Arabidopsis [174], rice [175], barley [176], and wheat [28,177]. HKT proteins transport monovalent cations through the electrophysiological gradient, and they have been divided into two subfamilies. The first subfamily in the first pore loop of the protein have a serine (S-G-G-G) and make them more selective for the Na^+^ ion than subfamily 2 members, which have glycine (G-G-G-G) in the first loop of the protein [178]. Laurie et al. [179] have shown that downregulation of *TaHKT2;1* (*TaHKT1*) in wheat increased the shoot fresh weight from 50% to 100% under 200 mM NaCl treatment and K^+^ deficiency. *TaHKT2;1* expressed in the root, including root hairs, perform as Na^+^/K^+^ symporters at low Na^+^ concentrations, and a Na^+^ uniporter at high Na^+^ concentrations [178,179]. One more gene induced by salt stress, and not involved in the SOS pathway, *TaSOS4*, encodes two pyridoxal kinases (PL), and their function is to convert vitamin B6 to pyridoxal 5′-phosphate (PLP) [180]. PLP in plants is the pivotal cofactor for various enzymes that are involved in biosynthesis of chlorophyll, ethylene, de novo sphingolipid, and metabolism of amino acids and carbohydrates [180]. In wheat, the gene encoding cytoplasmic pyridoxal kinase *TaSOS4* was identified with the amino acid sequence that is 78% identical to Arabidopsis *AtSOS4* [180].

### 5.4. The Genes Associated with Reduce Reactive Oxygen Species (ROS)

The plant response to oxidative stress start with producing antioxidant enzymes and a non-enzymatic antioxidant, which reduces the harmful effect of reactive oxygen species (ROS) [181]. Ten genes involved in the ROS are reported in Appendix A, with their relative reference, accession number, dimension (bp), annotation, function, and primers usable to amplify. The main enzymatic antioxidants involved in the scavenging of ROS are superoxide dismutase (SOD), catalase (CAT), ascorbate peroxidase (APX), glutathione reductase (GR), glutathione peroxidase (GPX), and peroxidase (POX).

Among the enzymes with antioxidant activity, the superoxide dismutase’s (SODs) serve as the first line of defense against abiotic stress and ROS [182]. Moreover, drought and salinity in *T. aestivum* affect differently mitochondrial *TaMn-SOD* (manganese superoxide dismutase) and cytosolic *TaCu/Zn-SOD* (Cu/Zn-superoxide dismutase). After the long-term effects of salt stress on *Triticum aestivum*, the total SOD activity was the highest in the salt-tolerant line chloroplastic fraction followed by mitochondrial, and the lowest in the cytosolic fraction [183,184].

Ascorbate and glutathione are two essential non-enzymatic compounds, reducing the harmful effect of oxidative stress via detoxification of hydrogen peroxide. Enzymes such as ascorbate peroxidase *TaAPX* and glutathione peroxidases *W69* and *W106* play key roles in maintaining the ascorbate and glutathione contents in plants [185]. Sairam and Saxena [186] determined that water-stress-tolerant wheat genotype PBW 175 had the highest ascorbate peroxidase, glutathione reductase, and peroxidase activity, as well had the lowest lipid peroxidation and highest membrane stability under water stress. Glutathione peroxidase (GPX) genes from wheat *W69* and *W106*, localized in chloroplasts, showed their peroxidase activity in vitro, and their overexpression, enhanced tolerance to salt, and H_2_O_2_ in transgenic Arabidopsis [187]. To decrease the harmful effects of H_2_O_2_, organic hydroperoxide, and lipid hydroperoxide, the GPX uses glutathione (GSH) or thioredoxin (Trx) as the reducing agents. Moreover, the overexpression of *Ta-sro1* (poly (ADP ribose) polymerase) in wheat promotes the activity of these ascorbate-GSH cycle enzymes and are involved in maintaining the genomic integrity, which allows plants to regulate redox homeostasis under salinity stress [188].

Peroxidases (PRXs) are also involved in various responses to abiotic stresses. *TaPRX-2A* gene, which is a member of the wheat class III peroxidase gene family, was recently cloned and characterized by Su et al. [189]. They detected an improved salt tolerance in wheat, with its main expression level located in the root tissues, with 1026 bp an open reading frame (ORF); furthermore, they showed that transgenic wheat plants with *TaPRX-2A*-overexpressed have also higher activities of other genes involved in the redox mechanisms. Moreover, wheat catalase *TaCAT* activities in leaves retain their water status, associated with resistance to drought, and preventing the grain yield components from being compromised [190].

Non-enzymatic antioxidant flavonoids belong to a large family of phenolic compounds, protecting plants against various environmental stresses [191]. The flavonoids’ biosynthesis is regulated genetically and controlled by key enzymes, one of them flavanone 3-hydroxylase, which is encoded by gene *F3H1*. Moreover, the *F3H1* belongs to the ‘early’ flavonoid synthesis gene family; its expression patterns under stress conditions can regulate the flavonoid biosynthesis pathway and produce antioxidants with high efficiency [192].

Plant 12-oxo-phytodienoic acid reductases (OPRs) that catalyze the reduction of double bonds in α,β-unsaturated aldehydes, and ketones, do not interact with jasmonate synthesis nor the jasmonate signaling pathway, but are dependent on the abscisic acid (ABA) regulation signaling pathway [193]. In *T. aestivum*, *TaOPR1* (oxophytodienoate reductase), enhanced salinity and drought tolerance, via promoting activity in an ABA-dependent pathway and Reactive Oxygen Species Scavenging [194]. Out of the 1347 bp full-length *TaOPR1* includes two untranslated regions at the 5′ and 3′ ends of 60-bp and 69-bp, respectively; while, the central bp open reading frame is of 1110-bp. The analyses with aneuploid stocks revealed its location on chromosome 2BS [193].

## 6. Conclusions

Wheat, like many other plants, reacts to hydric stress, activating several mechanisms, and these mechanisms involve several genes. The response could vary in accord with (i) the hydric stress typology (drought *versus* salinity); (ii) the plant stage; and (iii) the stress intensity. Knowledge and the understanding of the mechanisms involved in stresses tolerance, together with the genes activated by the stress, help the selection and use of plants suitable for cultivation in location with water scarcity (quantity and quality). This review aims to be a reference to understand these mechanisms and to have a clear list of the most important genes involved in conferring plant tolerance to hydric stresses. All the information here present are useful to properly run breeding programs to face climatic change either using classical breeding, helped by the specific molecular markers, or by using advanced biotechnological tools, able to transfer specific genes from a species to another.

## Figures and Tables

**Figure 1 ijms-22-06378-f001:**
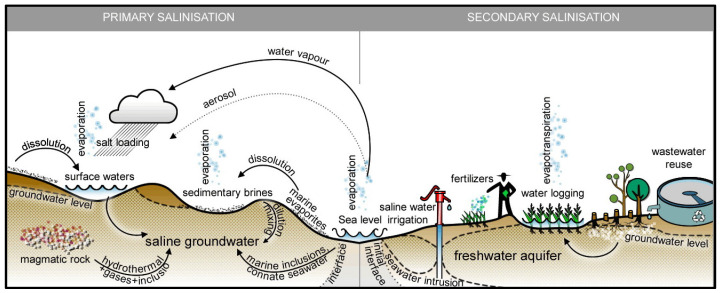
Primary and secondary soil salinity mechanisms (Reprinted with permission from ref. [12]. 2016, Elsevier).

**Figure 2 ijms-22-06378-f002:**
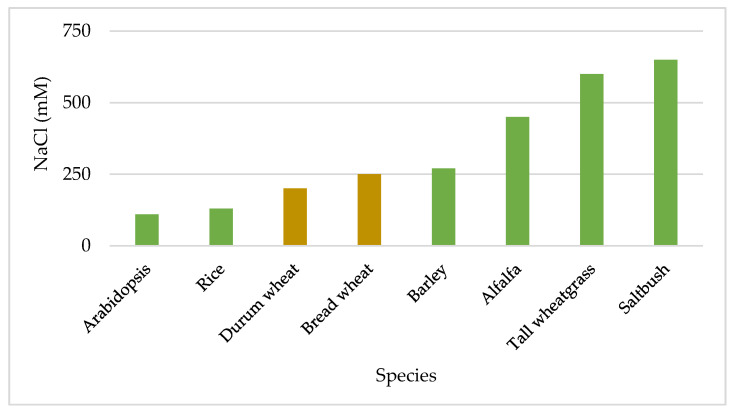
Salt tolerance differences among various species, expressed as NaCl concentration to inhibit dry matter production [21].

**Figure 3 ijms-22-06378-f003:**
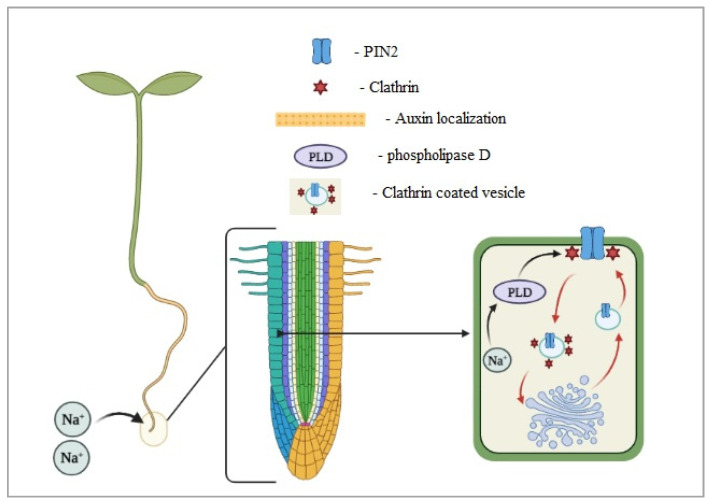
Salt-activated phospholipase D (PLD) induces clathrin-mediated endocytosis of PIN2, which allows reorganizing the auxin and roots’ growing direction (modified from Galvan-Ampudia et al. [72]).

**Figure 4 ijms-22-06378-f004:**
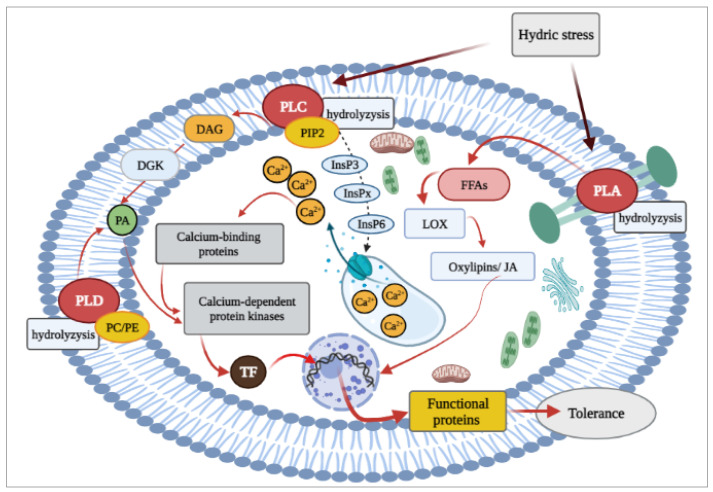
The general scheme of plant cell response to hydric stress in the lipid-dependent pathway: signal perception, transduction, and activation of the target genes. ABA—abscisic acid, PLD—phospholipases D, PC—phosphatidylcholine, PE—phosphatidylethanolamine, PA—phosphatidic acid, PLC—phospholipase C, PIP2—phosphatidylinositol bisphosphate, IP3—inositol triphosphate, DAG—diacylglycerol, DGK—diacylglycerol kinase, PLA—phospholipase A, FFAs—free fatty acids, LOX—lipoxygenase pathway, JA—jasmonic acid, TF—transcription factors.

**Figure 5 ijms-22-06378-f005:**
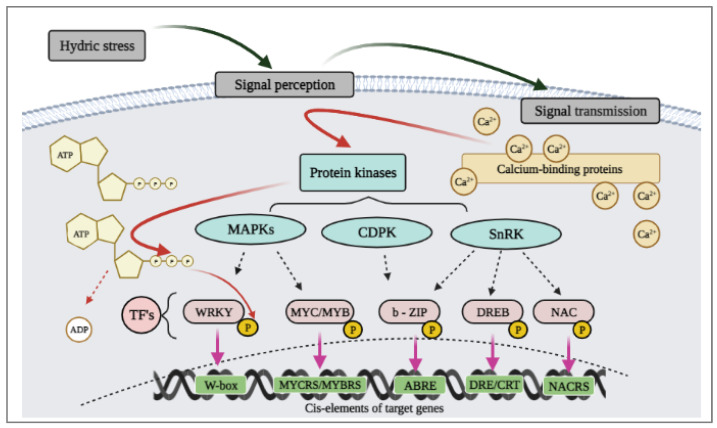
The general scheme for the activation of transcription factors by different protein kinase families in response to hydric stress. TFs—transcription factors, MAPKs—mitogen-activated protein kinases, CDPK—calcium-dependent protein kinases, SnRK2—serine-threonine kinases.

**Table 2 ijms-22-06378-t002:** Plant salinity-tolerance mechanisms, ordered by processes and their relevance for the three components of salinity tolerance [21].

		Osmotic Stress	Ionic Stress		
Process	Candidate Genes	Osmotic Tolerance	Na^+^ Excluding	Tissue Tolerance	References
Signaling	*SOS3*, *SnRKs*	Signaling regulation	Activation of ion antiporter	Regulation of vacuolar loading	[77]
Photosynthesis	*ERA1*, *PP2C*, *AAPK*, *PKS3*	Stomatal closure regulation	Protection of chloroplast from ion toxicity	Delay Na^+^ toxicity effect in chloroplast	[78,79]
Accumulation of Na^+^ in shoots	*HKT*, *SOS1*	-	Decreasing long distance transport of Na^+^	Decreasing energy used on Na^+^ exclusion	[80,81,82]
Accumulation of Na^+^in vacuoles	*NHX*, *AVP*	-	Increased sequestration of Na^+^ into root vacuoles	Increased sequestration of Na^+^ into leaf vacuoles	[80,83]
Accumulation of organic solutes	*P5CS*, *OTS*, *MT1D*, *M6PR*, *S6PDH*, *IMT1*	Increasing osmotic adjustment	Reduction of Na^+^ accumulation	Accumulation of organic solutes in cytoplasm	[82,84]

**Table 3 ijms-22-06378-t003:** Timing of the plant’s response to salinity after the stress was imposed. The effects on a salt-tolerant plants are fundamentally identical to those due to soil water deficit (Reprinted with permission from ref. [99]. 2002, John Wiley and Sons).

Time	Water Stress Effect (Salt-Tolerant Plants)	Salt-Specific Effects Salt-Sensitive Plants
Minutes	Immediate reduction in leaf and root elongation rate and then rapid partial recovery	
Hours	Constant but reduced rate of leaf and root elongation	
Days	Leaf growth more affected than root growth; Reduced rate of leaf emergence	Visible injury in the oldest leaf
Weeks	Reduced the final size of the leaves and/or the number of side shoots	Death of older leaves
Months	Altered flowering time, reduced seed production	Younger leaves dead, plants may die before the seed matures

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
