# Peer review of "One Hundred Candidate Genes and Their Roles in Drought and Salt Tolerance in Wheat"

_ijms, 2021, doi:10.3390/ijms22126378_

Round 1

Reviewer 1 Report

The authors described plant mechanism of salt and drought in wheat.

In current review, they included several aspects of wheat in response to salt and drought stresses. 

First of all, I cannot easily catch of the main objectives of this manuscript. In addition, overall introduction part was not clear to explain why this review should be important at this moment.

Accumulating data already demonstrated any specific mechanisms, hormonal pathways, genes, QTLs, and signaling pathways in wheat species.

Moreover, any strong emphasizing issues, perspective, and future directions did not included in this paper.

I feel this review is just repeatable from other papers. Therefore, unfortunately, I cannot recommend this paper. 

Author Response

Answers to Reviewer 1

First of all I would like to apologies with reviewers since, in order to avoid too many track changes we did not adjusted the Tables and Caps which are share among two pages. Of course they will be moved properly to avoid this.

The authors described plant mechanism of salt and drought in wheat.

In current review, they included several aspects of wheat in response to salt and drought stresses. 

First of all, I cannot easily catch of the main objectives of this manuscript. In addition, overall introduction part was not clear to explain why this review should be important at this moment.

To make more easily catch the main objectives of this manuscript, other than in the abstract, the aim is added at the end of the first sections modifying the previous last-paragraph and adding another one (rows 98-100 and 141-146). Moreover to make the review aims as well as its unicity more evident we also change the paper Title.

Accumulating data already demonstrated any specific mechanisms, hormonal pathways, genes, QTLs, and signaling pathways in wheat species.

Moreover, any strong emphasizing issues, perspective, and future directions did not included in this paper.

Generally a review does not indicate future directions. Nevertheless, we add a sentence at the end of the conclusion to answer the reviewer point (rows 1133-1137).

I feel this review is just repeatable from other papers.

Of course any review somehow repeats other papers, but we disagree with the reviewer since the unicity of present review are several. In particularly, it put in a single paper all the necessary aspects to “solve the problem” from a breeding point of view. It provides the genetic resources useful to transfer genes to wheat germplasm. It focuses on the morpho-physiological aspect important to confer tolerance. It provides in the same place updated information on the main genes involved in the tolerance to drought and salt stresses.

Therefore, unfortunately, I cannot recommend this paper. 

Reviewer 2 Report

In this review, the authors describe current situation and studies related to hydric stress in plants and collect information about wheat genes. I have some comments for the improvement.

  1. Concept is unclear in the third paragraph in section 1. “drought and salinity are the main ..factors…. However, ….by other factors…”(L.47-50). These sentences should be paired but the focus of following sentences are not logically straight forward or the definition of `drought´ is unclear.
  2. The definition of `Climate change´ (L.65) is unclear. Temperature change `is´ the climate change, which cannot have impact on itself.
  3. A reference/references should be cited about WUE (L.80)
  4. It is difficult to understand the logic on “the formers give information that is more useful about single genes for pyramiding useful genes” (L.116)
  5. “As a results…” (L.128). I think that the conclusion “resistance is present in D” came from the results that AABBDD is more tolerant than AABB. The logic is opposite.
  6. What is the definition of “single gene mechanism” (L.134)
  7. It is difficult for me to understand the sentence “Unfortunately, the amphiploid mix….characters.” (L.151-152)
  8. It is difficult to understand the logic on “Also the A genome….AABB)” (L.169-171). A reference/references should also be cited.
  9. It is difficult for me to understand the sentence “Farooq and coworkers,…...” (L.174-177)
  10. What are altered in PIN2 messenger RNA and sos genes? (L.244)
  11. More explanation should be in Fig.3 legend.
  12. References should be shown in Table2.
  13. “Son of Sevenless” (L.319) is a Drosophila gene, which has no homology to plant SOS. It should be Salt Overly Sensitive as written in L.244 and 644.
  14. Ref#91 mentions nothing on NO. (L.367)
  15. IP3 (L.417 and Fig.4). The authors should read Krinke, et al. (2006) Inositol trisphosphate receptor in higher plants: is it real? J. Exp. Bot. 58: 361–376
  16. “Ca2+ EF-hand binding proteins” (L.456) It sounds weird. “EF-hand protein” or “Ca2+ binding EF-hand proteins”.
  17. SnRK2s are not calcium-binding proteins (L.475-480)
  18. SnRK (Fig5 legend and L.520) is “SNF1-related protein kinase”
  19. “with the ABA-responsive cis-element ABRE” (L.531) What does this “with” mean?
  20. “more important” (L545) Why are they more important?

  1. I think that the followings are mistakes although I am not an English native speaker.

successful > successfully (L.109),  stomas > stomata (L.112), This because > This is because (L.136), looking > looked (L.158), than > that (L.195), cells plants > plant cells (L.229), demonstrated > is demonstrated (L.432), However > (remove it) (L.510), overexpressed > are induced (L.594), induced > activated (L.652), However > (remove it) (L.712),

Author Response

Answers to Reviewer 2

First of all I would like to apologies with reviewers since, in order to avoid too many track changes we did not adjusted the Tables and Caps which are share among two pages. Of course they will be moved properly to avoid this.

In this review, the authors describe current situation and studies related to hydric stress in plants and collect information about wheat genes. I have some comments for the improvement.

  1. Concept is unclear in the third paragraph in section 1. “drought and salinity are the main ..factors…. However, ….by other factors…”(L.47-50). These sentences should be paired but the focus of following sentences are not logically straight forward or the definition of `drought´ is unclear.

Thanks for the comment we clarify the concept listing also some of the other consequences of drought on plant.

It is now « However, other than grain production water scarcity affect also the growth rate, leaf size, stem extension, root proliferation, susceptibility to disease, plant color, etc. [9,10]»

  1. The definition of `Climate change´ (L.65) is unclear. Temperature change `is´ the climate change, which cannot have impact on itself.

We agree with the rounding definition, so we modify to make it clearer.

It is now « Climate change is the increase in temperatures often associated with rainfall decrease, but it leads to more extreme and unpredictable events. So the needs to have a system protecting from erratic clime becoming important all around the world.»

  1. A reference/references should be cited about WUE (L.80)

As requested we added a recent review on WUE.

It is: Hatfield J. L., Dold C. Water-Use Efficiency: Advances and Challenges in a Changing Climate. Frontiers in Plant Science (10) 103 2019. DOI=10.3389/fpls.2019.00103

  1. It is difficult to understand the logic on “the formers give information that is more useful about single genes for pyramiding useful genes” (L.116)

Thanks. The sentence is reformulated.

It is: «While the formers give information about single genes; information useful for gene pyramiding and to understand the physiological mechanism involved in the tolerance.»

  1.  “As a results…” (L.128). I think that the conclusion “resistance is present in D” came from the results that AABBDD is more tolerant than AABB. The logic is opposite.

We are sorry but probably we don’t understand the review point. We agree with Reviewer that “the conclusion “resistance is present in D” came from the results that AABBDD is more tolerant than AABB”. In fact, bread wheat having D accumulates less sodium than durum wheat without D genome.

  1. What is the definition of “single gene mechanism” (L.134)

We change it into “single gene action”. It is more important a single gene than not a complexity of several genes.

  1. It is difficult for me to understand the sentence “Unfortunately, the amphiploid mix….characters.” (L.151-152)

We are sorry about that. Since the amphiploids derived from interspecific crosses have chromosomes of both species, usually the wild species has useful genes for tolerance, but not for production (see also previous rows 134-138).

We changed “mix” with “combine”, and “donors’” with “donors’”; probably more appropriate.

  1. It is difficult to understand the logic on “Also the A genome….AABB)” (L.169-171). A reference/references should also be cited.

The logic of this point is similar to the one reported on the previous point 5. The referece [27] (i.e. Colmer, Timothy D., Timothy J. Flowers, and Rana Munns. "Use of wild relatives to improve salt tolerance in wheat." Journal of experimental botany 57.5 (2006): 1059-1078.) was added.

  1. It is difficult for me to understand the sentence “Farooq and coworkers,…...” (L.174-177)

We apologize, inserting Table 1 the sentence was wrongly allocated to the previous paragraph. It is now moved in the paragraph after Table 1.

  1. What are altered in PIN2 messenger RNA and sos genes? (L.244)

It is altered the PIN2 abundance and polar distribution. SOS mediate the decrease of PIN2 messenger RNA in salinity-induced modification of gravitropic response in Arabidopsis roots. We did not modify the text since reader could go to the cited reference to have more details.

  1. More explanation should be in Fig.3 legend.

Thank you, explanation about Fig. 3 was detailed in the manuscript.

  1. References should be shown in Table2.

References are inserted as requested,

  1.  “Son of Sevenless” (L.319) is a Drosophila gene, which has no homology to plant SOS. It should be Salt Overly Sensitive as written in L.244 and 644.

Thank you so much. We corrected it.

  1. Ref#91 mentions nothing on NO. (L.367)

We apologize the wrong references was mentioned, we correct it.

  1. IP3 (L.417 and Fig.4). The authors should read Krinke, et al. (2006) Inositol trisphosphate receptor in higher plants: is it real? J. Exp. Bot. 58: 361–376

Thank you for your suggestion, article was very interesting. To our manuscript we added references, which say that IP3 involved in calcium ions release under stress conditions, as shown in Fig4.

Lin, W. H., Rui, Y. E., Hui, M. A., Xu, Z. H., & Xue, H. W. (2004). DNA chip-based expression profile analysis indicates involvement of the phosphatidylinositol signaling pathway in multiple plant responses to hormone and abiotic treatments. Cell research14(1), 34-45.

  1.  “Ca2+ EF-hand binding proteins” (L.456) It sounds weird. “EF-hand protein” or “Ca2+ binding EF-hand proteins”.

Thank you, we corrected it into “calcium-binding EF-hand proteins”

  1. SnRK2s are not calcium-binding proteins (L.475-480)

Thank you, we reword it to made clearer that it is a protein kinase.

  1. SnRK (Fig5 legend and L.520) is “SNF1-related protein kinase”

Thank you. SnRK was corrected to SnRK2, which is SNF1-type serine-threonine protein kinase.

  1.  “with the ABA-responsive cis-element ABRE” (L.531) What does this “with” mean?“

Thank you, and sorry for misunderstanding. It was corrected.

  1. more important” (L545) Why are they more important?

Thank you, it was corrected.

  1. I think that the followings are mistakes although I am not an English native speaker.

Thanks for the suggestions

successful > successfully (L.109),

Changed

  stomas > stomata (L.112),

Changed rewording

 This because > This is because (L.136),

Changed

 looking > looked (L.158),

Inserted the coma that was missing

than > that (L.195),

Re phrased into: “Thinopyrum bessarabicum (Savul and Rayss) Á. Löve (JJ syn, EbEb; 2n=14) could be much more convenient than polyploids for wheat cytogenetic manipulations”

 cells plants > plant cells (L.229),

Changed

 demonstrated > is demonstrated (L.432)

Changed

However > (remove it) (L.510),

Deleted

 overexpressed > are induced (L.594),

Changed

 induced > activated (L.652),

Changed

 However > (remove it) (L.712),

Deleted

Round 2

Reviewer 1 Report

I do not have any comments and suggestions for authors.

Author Response

No comments for Rev 1

Reviewer 2 Report

If the authors think that IP3 is reasonable in Fig.4, they should discuss it spending one section. Otherwise, audience may just put this review into a trash box. Other points are below.

5.“As a results…” (L.128). I think that the conclusion “resistance is present in D” came from the results that AABBDD is more tolerant than AABB. The logic is opposite.

We are sorry but probably we don’t understand the review point. We agree with Reviewer that “the conclusion “resistance is present in D” came from the results that AABBDD is more tolerant than AABB”. In fact, bread wheat having D accumulates less sodium than durum wheat without D genome.

I mean, you should remove the terms “as a results”.

Because AABBDD is more tolerant than AABB, “resistance is present in D” was concluded. That is scientific logic. However, the current logic is “because resistance is present in D, AABBDD is more tolerant than AABB”. That is a kind of religious description.(First, someone set the resistance in D. That is the reason AABBDD is more tolerant.)

6.What is the definition of “single gene mechanism” (L.134)

We change it into “single gene action”. It is more important a single gene than not a complexity of several genes.

It is still unclear for me. The authors should write more precisely. For example, "Useful source of genes for tolerance are in the wild relatives, in which a single gene, not a complexity of several genes, often confer the increase of yield under stress (see below)."

8.It is difficult to understand the logic on “Also the A genome….AABB)” (L.169-171). A reference/references should also be cited.

The logic of this point is similar to the one reported on the previous point 5. The referece [27] (i.e. Colmer, Timothy D., Timothy J. Flowers, and Rana Munns. "Use of wild relatives to improve salt tolerance in wheat." Journal of experimental botany 57.5 (2006): 1059-1078.) was added.

The fact that AA is more tolerant than AABB indicates that B has a factor inducing less Na+ exclusion but not that A has the factor. A is present in both.

10.What are altered in PIN2 messenger RNA and sos genes? (L.244)

It is altered the PIN2 abundance and polar distribution. SOS mediate the decrease of PIN2 messenger RNA in salinity-induced modification of gravitropic response in Arabidopsis roots. We did not modify the text since reader could go to the cited reference to have more details.

It is confusing. With “altering PIN2 messenger RNA” I first think splice variants or secondary structure of RNA. “altering the PIN2 abundance and polar distribution” is better.

Moreover, the phrase “altering salt overly sensitive genes” suggests genome recombination or so. If the authors want to mention “SOS mediate the decrease of PIN2 messenger RNA in salinity-induced modification of gravitropic response in Arabidopsis roots.”, the current sentence is incorrect.

15. IP3 (L.417 and Fig.4). The authors should read Krinke, et al. (2006) Inositol trisphosphate receptor in higher plants: is it real? Exp. Bot. 58: 361–376

Thank you for your suggestion, article was very interesting. To our manuscript we added references, which say that IP3 involved in calcium ions release under stress conditions, as shown in Fig4.

Lin, W. H., Rui, Y. E., Hui, M. A., Xu, Z. H., & Xue, H. W. (2004). DNA chip-based expression profile analysis indicates involvement of the phosphatidylinositol signaling pathway in multiple plant responses to hormone and abiotic treatments. Cell research, 14(1), 34-45.

Again, I suggest you to read Krinke et al. in 2006. Your cited literature was published in 2004 and Krinke et al. discussed why the literature in 2004 is not valid anymore. If you think Lin et al. (2004) is more reasonable, you should write a review on that instead of this.  

19. “with the ABA-responsive cis-element ABRE” (L.531) What does this “with” mean?“

Thank you, and sorry for misunderstanding. It was corrected.

Another large and most diverse TF superfamily belongs to the important subfamily of basic leucine zipper (bZIP) or AREB/ABF TFs, which carrying a highly conserved bZIP domain. These TFs activate the genes, especially responsible for drought tolerance and have the ABA-responsive cis-element ABRE, like numerous LEA family genes, such as cold regulated (COR), responsive to dehydration (RD), early responsive to dehydration (ERD), and responsive to ABA (RAB) [129]. (L.817-822)

It is unclear. Reconsider the structure of the sentence. What does “or” mean? It is unclear which elements are parallel.

Author Response

Reviewer2

If the authors think that IP3 is reasonable in Fig.4, they should discuss it spending one section. Otherwise, audience may just put this review into a trash box. Other points are below.

A paragraph as short summary about lipid-dependent signaling pathway is already in our paper and adding new section will be out of the paper scope. Figure 4 was modified by using recent updates in this field [108,109]. Please see also point 15.

5.“As a results…” (L.128). I think that the conclusion “resistance is present in D” came from the results that AABBDD is more tolerant than AABB. The logic is opposite.

We are sorry but probably we don’t understand the review point. We agree with Reviewer that “the conclusion “resistance is present in D” came from the results that AABBDD is more tolerant than AABB”. In fact, bread wheat having D accumulates less sodium than durum wheat without D genome.

I mean, you should remove the terms “as a results”.

Because AABBDD is more tolerant than AABB, “resistance is present in D” was concluded. That is scientific logic. However, the current logic is “because resistance is present in D, AABBDD is more tolerant than AABB”. That is a kind of religious description.(First, someone set the resistance in D. That is the reason AABBDD is more tolerant.)

We removed “As a results”.

6.What is the definition of “single gene mechanism” (L.134)

We change it into “single gene action”. It is more important a single gene than not a complexity of several genes.

It is still unclear for me. The authors should write more precisely. For example, "Useful source of genes for tolerance are in the wild relatives, in which a single gene, not a complexity of several genes, often confer the increase of yield under stress (see below)."

We modified the sentence to make it clearer.

8.It is difficult to understand the logic on “Also the A genome….AABB)” (L.169-171). A reference/references should also be cited.

The logic of this point is similar to the one reported on the previous point 5. The referece [27] (i.e. Colmer, Timothy D., Timothy J. Flowers, and Rana Munns. "Use of wild relatives to improve salt tolerance in wheat." Journal of experimental botany 57.5 (2006): 1059-1078.) was added.

The fact that AA is more tolerant than AABB indicates that B has a factor inducing less Na+ exclusion but not that A has the factor. A is present in both.

We apologize, reviewer is right. We added the possible justification of this.

10.What are altered in PIN2 messenger RNA and sos genes? (L.244)

It is altered the PIN2 abundance and polar distribution. SOS mediate the decrease of PIN2 messenger RNA in salinity-induced modification of gravitropic response in Arabidopsis roots. We did not modify the text since reader could go to the cited reference to have more details.

It is confusing. With “altering PIN2 messenger RNA” I first think splice variants or secondary structure of RNA. “altering the PIN2 abundance and polar distribution” is better.

Moreover, the phrase “altering salt overly sensitive genes” suggests genome recombination or so. If the authors want to mention “SOS mediate the decrease of PIN2 messenger RNA in salinity-induced modification of gravitropic response in Arabidopsis roots.”, the current sentence is incorrect.

Sentence was changed as suggested by Academic Editor.

  1. IP3 (L.417 and Fig.4). The authors should read Krinke, et al. (2006) Inositol trisphosphate receptor in higher plants: is it real? Exp. Bot. 58: 361–376

Thank you for your suggestion, article was very interesting. To our manuscript we added references, which say that IP3 involved in calcium ions release under stress conditions, as shown in Fig4.

Lin, W. H., Rui, Y. E., Hui, M. A., Xu, Z. H., & Xue, H. W. (2004). DNA chip-based expression profile analysis indicates involvement of the phosphatidylinositol signaling pathway in multiple plant responses to hormone and abiotic treatments. Cell research, 14(1), 34-45.

Again, I suggest you to read Krinke et al. in 2006. Your cited literature was published in 2004 and Krinke et al. discussed why the literature in 2004 is not valid anymore. If you think Lin et al. (2004) is more reasonable, you should write a review on that instead of this.  

We agree that Lin et al. (2004) wasn’t the best choice for reference.

However, if there is no direct genetic or molecular evidence for the existence of IP3 receptors in plants, it does not mean that it is not exist only because it does not share sequence homology with the analogous animal proteins. Two references were added to the manuscript [108,109], which represent recent updates in this field.

Gillaspy, G. E. (2011). The cellular language of myo‐inositol signaling. New Phytologist, 192(4), 823-839

Munnik, T.; Vermeer, J.E.M. Osmotic stress-induced phosphoinositide and inositol phosphate signaling in plants. Plant, Cell Environ. 2010, 33(4), 655-669. doi:10.1111/j.1365-3040.2009.02097.x.

Sentence was then modified into:

“The hydrolysis of phosphatidylinositol bisphosphate (PIP2) by phospholipase C produces diacylglycerol (DAG) and inositol triphosphate (InsP3). In animals IP3 is responsible to release calcium, while in plants this function is related to IP3 and his derivative IP6 (Figure 4).”

  1. “with the ABA-responsive cis-element ABRE” (L.531) What does this “with” mean?“

Thank you, and sorry for misunderstanding. It was corrected.

Another large and most diverse TF superfamily belongs to the important subfamily of basic leucine zipper (bZIP) or AREB/ABF TFs, which carrying a highly conserved bZIP domain. These TFs activate the genes, especially responsible for drought tolerance and have the ABA-responsive cis-element ABRE, like numerous LEA family genes, such as cold regulated (COR), responsive to dehydration (RD), early responsive to dehydration (ERD), and responsive to ABA (RAB) [129]. (L.817-822)

It is unclear. Reconsider the structure of the sentence. What does “or” mean? It is unclear which elements are parallel.

Sentence was reformulated into “The bZIP proteins belongs to the most large and diverse TF superfamily, classified into 14 subgroups, and carrying a highly conserved bZIP domain, which include a basic region and a leucine zipper. Members of bZIP family that carry ABA response element binding factors (AREBs) belong to subgroup-A, and are involved in abscisic acid signaling respond. These transcription factors regulate expression of genes responsible for drought tolerance and have ABA-responsive cis-element (ABRE), as numerous of the LEA family……..”

Round 3

Reviewer 2 Report

I do not have any further comments.